# VARIATIONAL BAYES GAUSSIAN SPLATTING

## ABSTRACT

Recently, 3D Gaussian Splatting has emerged as a promising approach for modeling 3D scenes using mixtures of Gaussians. The predominant optimization method for these models relies on backpropagating gradients through a differentiable rendering pipeline, which struggles with catastrophic forgetting when dealing with continuous streams of data. To address this limitation, we propose Variational Bayes Gaussian Splatting (VBGS), a novel approach that frames training a Gaussian splat as variational inference over model parameters. By leveraging the conjugacy properties of multivariate Gaussians, we derive a closed-form variational update rule, allowing efficient updates from partial, sequential observations without the need for replay buffers. Our experiments show that VBGS not only matches state-of-the-art performance on static datasets, but also enables continual learning from sequentially streamed 2D and 3D data, drastically improving performance in this setting.

## 1 INTRODUCTION

Representing 3D scene information is a long-standing challenge for robotics and computer vision (Özyeşil et al., 2017). A recent breakthrough in this domain relies on representing the scene as a radiance field, i.e., using Neural Radiance Fields (NeRF) (Mildenhall et al., 2020). Recently, 3D Gaussian Splatting (3DGS) (Kerbl et al., 2023) has demonstrated the effectiveness of mixture models for 3D scene representation, as highlighted by a surge of subsequent research (see Chen and Wang (2024) for a survey). This approach leverages the ability of Gaussians to represent physical space as a collection of ellipsoids. The dominant method for optimizing these models involves backpropagating gradients through a differentiable renderer with respect to the parameters of the mixture model.

While effective, 3DGS – and gradient-based methods more broadly – faces critical challenges in continual learning scenarios. Many real-world applications, such as autonomous navigation, simultaneous localization and mapping (SLAM), and robotics involve data that is continuously streamed and must be processed sequentially. In such settings, gradient-based optimization methods are prone to catastrophic forgetting (French, 1999), leading to performance degradation as new data overwrites old knowledge. To mitigate this, replay buffers are often employed to retain and retrain on older data (Matsuki et al., 2024), which can be computationally expensive and memory intensive.

To overcome these challenges, we propose Variational Bayes Gaussian Splatting (VBGS), casting Gaussian Splatting as a variational inference problem over the parameters of a generative mixture model, enabling a closed-form update rule (Blei et al., 2017). Our approach inherently supports continual learning through iterative updates that are naturally accumulative, eliminating the need for replay buffers.

Our contributions are summarized as follows:

- We propose a generative model of mixtures of Gaussians for representing spatial 2D (image) and 3D (point cloud) data and derive closed-form variational Bayes update rules.

- We show that VBGS enables continual learning without catastrophic forgetting, and VBGS models achieve a good model fit with only a single parameter update step.

- We demonstrate and benchmark our approach on both 2D image (TinyImageNet (Le and Yang, 2015)) and 3D point cloud (Blender 3D models (Mildenhall et al., 2020) and Habitat scenes (Savva et al., 2019)) datasets.

Our approach offers a robust alternative to gradient-based optimization by allowing continual, efficient updates from sequential data, making it well-suited for real-world applications.

## 2 RELATED WORK

**3D Gaussian Splatting (3DGS)**: Recently, 3DGS (Kerbl et al., 2023) introduced a novel approach for learning the structure of the world directly from image data by representing the radiance field as a mixture of Gaussians. This method optimizes the parameters of the Gaussian mixture model by backpropagating gradients through a differentiable rendering pipeline. The Gaussians serve as ellipsoid shape primitives for scene reconstruction, with each component associated with deterministic, learned features such as opacity and color. This approach has triggered significant advances in novel view synthesis and sparked interest in many applications across robotics, virtual/augmented reality, and interactive media (Fei et al., 2024; Chen and Wang, 2024).

While our method also represents scenes as a mixture of Gaussians, it diverges from 3DGS by modeling a distribution over all features and parameters. Rather than relying on gradient-based optimization through differentiable rendering, we frame the learning process as variational inference. This approach allows us to perform continual learning through closed-form updates, enabling more efficient and robust handling of sequential data.

**Image representations**: Gaussian mixture models have also been used for representing image data, particularly for data efficiency and compression (Verhack et al., 2020). Steered Mixture-of-Experts (SMoE) models, for instance, typically estimate parameters using the Expectation-Maximization (EM) algorithm, although some work has explored gradient descent optimization (Bochinski et al., 2018).

Our approach can be seen as an SMoE model on 2D image data, but again, instead of relying on EM or gradient descent, we propose variational Bayes parameter updates for more robust and efficient parameter estimation.

**Continual learning** aims to develop models that can adapt to new tasks from a continuous datastream without forgetting previously learned knowledge (Wang et al., 2024). Backpropagation-based methods, such as 3DGS, face significant challenges in this area due to catastrophic forgetting (French, 1999), where the model's performance on prior data deteriorates as it trains on new. This issue is particularly evident in real-world scenarios, such as localization and mapping, where data is streamed (Matsuki et al., 2024; Keetha et al., 2024). The common mitigation strategy involves maintaining a replay buffer of frames to revisit past data during updates (Sucar et al., 2021; Fu et al., 2024; Zhipeng Cai, 2023).

In contrast, Bayesian approaches offer a more elegant solution for online learning, especially when dealing with conjugate models, allowing for exact Bayesian inference in a sequential setting (Jones et al., 2024). We leverage this property to update Gaussian splats continuously as new data arrives, eliminating the need for replay buffers.

## 3 METHOD

VBGS relies on the conjugate properties of exponential family distributions. This is well suited for variational inference, as we can derive closed-form update rules for inferring the (approximate) posterior. We first write down the functional form of these distributions and then describe the particular generative model for representing space and color.

If the likelihood distribution is part of the exponential family, it can be written down as:

$$p(x|\theta) = \phi(x) \exp\left(\theta \cdot T(x) - A(\theta)\right), \tag{1}$$

where $\theta$ are the natural parameters of the distribution, $T(x)$ is the sufficient statistic, $A(\theta)$ is the log partition function, and $\phi(x)$ is the measure function. This likelihood has a conjugate prior of the following form:

$$p(\theta|\eta_0, \nu_0) = \frac{1}{Z(\eta_0, \nu_0)} \exp\left(\eta_0 \cdot \theta - \nu_0 \cdot A(\theta)\right), \tag{2}$$

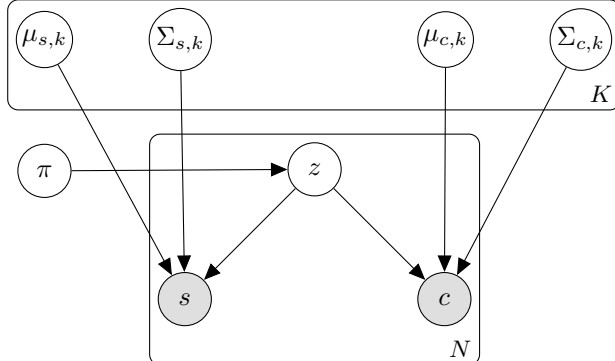

Figure 1: **The Generative Model**: There are $K$ components in the mixture model, and $N$ observed data points, which are composed of a spatial vector $s$, and feature vector $c$. The parameters of the distributions of component $k$ ($\mu_k, \Sigma_k$) that generate $s$ and $c$ are random variables, which influence $s$ and $c$ respectively. $z$ is the associated mixture component for a given data point, dependent on the categorical parameters $\pi$. White and gray circles denote unobserved and observed variables respectively.

where $Z(\eta_0, \nu_0)$ is the normalizing term. This distribution is parameterized by its natural parameters $(\eta_0, \nu_0)$. The posterior is then of the same functional form as the prior, whose natural parameters can be calculated as a function of the sufficient statistics of the data and the natural parameters of the prior, i.e. $(\eta_0 + \sum_x T(x), \nu_0 + \sum_x 1)$ (Murphy, 2013).

### 3.1 THE GENERATIVE MODEL

The generative model considered is a mixture model with $K$ components, each characterized by two conditionally independent modalities: spatial position ($s$) and color ($c$). For 2D images, $s$ represents the pixel location in terms of row and column coordinates ($s \in \mathbb{R}^2$), while for 3D data, it corresponds to Cartesian coordinates ($s \in \mathbb{R}^3$). The color component represents RGB values ($c \in \mathbb{R}^3$) for both 2D and 3D data. Both $s_k$, and $c_k$ are modeled as multivariate Normal distributions with parameters $(\mu_{s,k}, \Sigma_{s,k})$, and $(\mu_{c,k}, \Sigma_{c,k})$ respectively, and the components have a mixture weight $z$. The full generative model is visualized as a Bayesian network in Figure 1 and is factorized as:

$$p(s, c, z, \mu_s, \Sigma_s, \mu_c, \Sigma_c, \pi) = \left( \prod_{n=1}^{N} p(s_n | z_n, \mu_s, \Sigma_s) p(c_n | z_n, \mu_c, \Sigma_c) p(z_n | \pi) \right) \tag{3}$$

$$\left( \prod_{k=1}^{K} p(\mu_{k,s}, \Sigma_{k,s}) p(\mu_{k,c}, \Sigma_{k,c}) \right) p(\pi). \tag{4}$$

The joint distribution in Equation (4) is factorized into two main components: one representing the likelihoods of $N$ data points $(s_n, c_n)$ given their assignments, and another representing the priors over the mixture model parameters. The mixture components over space $p(s | z, \mu_s, \Sigma_s)$ are conditionally independent from the component over color $p(c | z, \mu_c, \Sigma_c)$, given mixture component $z$. The following distributions parameterize these random variables:

$$z_n \sim \text{Cat}(\pi) \tag{5} \qquad c_n | z_n = k \sim \text{MVN}(\mu_{k,c}, \Sigma_{k,c}) \tag{7}$$
$$s_n | z_n = k \sim \text{MVN}(\mu_{k,s}, \Sigma_{k,s}) \tag{6}$$

The parameters of these distributions are also modeled as random variables. Treating the parameters as latent random variables allows us to cast learning as inference using the appropriate conjugate priors. The conjugate prior to a multivariate Normal (MVN) is a Normal Inverse Wishart (NIW) distribution, and the Dirichlet distribution is the conjugate prior to a categorical (Cat) distribution:

$$\mu_{k,s}, \Sigma_{k,s} \sim \text{NIW}(m_{0,s}, \kappa_{0,s}, V_{0,s}, n_{0,s}) \tag{8}$$

$$\mu_{k,c}, \Sigma_{k,c} \sim \text{NIW}(m_{0,c}, \kappa_{0,c}, V_{0,c}, n_{0,c}) \tag{9}$$

$$\pi \sim \text{Dirichlet}(\alpha_0), \tag{10}$$

See Appendix A.1 for a table with the values used for the hyperparameters.

To estimate the parameters, we infer their posterior distribution. However, as computing this posterior is intractable, we resort to variational inference (Jordan et al., 1998). We use a mean-field approximation to make inference tractable, which assumes that the variational posterior factorizes across the latent variables, as otherwise the interaction between the various Gaussian components makes the inference process grow combinatorially. This factorization allows for efficient coordinate ascent updates for each variable separately. Specifically, the variational distribution $q$ is decomposed as:

$$q(z, \mu_s, \Sigma_s, \mu_c, \Sigma_c, \pi) = \left( \prod_{n=1}^{N} q(z_n) \right) \left( \prod_{k=1}^{K} q(\mu_{k,c}, \Sigma_{k,c}) \right) \left( \prod_{k=1}^{K} q(\mu_{k,s}, \Sigma_{k,s}) \right) q(\pi), \tag{11}$$

We select the distributions of the approximate posteriors to be from the same family as their corresponding priors. For the color variable $c$, we model the mean by a Normal distribution but keep the covariance fixed using a Delta distribution. This assures that the mixture components commit to a particular color and do not blend multiple neighboring colors in a single component.

$$q(z_n) = \text{Cat}(\gamma_n) \tag{12}$$

$$q(\mu_{k,s}, \Sigma_{k,s}) = \text{NIW}(m_{t,s}, \kappa_{t,s}, V_{t,s}, n_{t,s}) \tag{13}$$

$$q(\pi) = \text{Dirichlet}(\alpha_t) \tag{14}$$

$$q(\Sigma_{k,c}) = \text{Delta}(\varepsilon I), \tag{15}$$

$$q(\mu_{k,c}) = \text{Normal}(m_{t,c}, \kappa_{t,c}^{-1} \varepsilon I) \tag{16}$$

where the subscript $t$ indicates the parameters at timestep $t$, and $\varepsilon$ is a chosen hyperparameter.

Images can be generated using the mixture model by computing the expected value of the color, conditioned on a spatial coordinate ($\mathbb{E}_{p(c|s)}[c]$). For 3D rendering, we use the renderer from Kerbl et al. (2023), where the spatial component is first projected onto the image plane using the camera parameters, and the estimated depth is used to deal with occlusion. For more details, see Appendix B.

## 3.2 Coordinate ascent variational inference

To estimate the posterior over the model parameters, we maximize the Evidence Lower Bound (ELBO) $\mathcal{L}$ with respect to the variational parameters (Jordan et al., 1998):

$$\mathcal{L} = \mathbb{D}_{\text{KL}}[q(z, \mu_s, \Sigma_s, \mu_c, \Sigma_c, \pi) \,||\, p(z, \mu_s, \Sigma_s, \mu_c, \Sigma_c, \pi | s, c)], \tag{17}$$

This is done using coordinate ascent variational inference (CAVI) (Beal, 2003; Bishop, 2006; Blei et al., 2017) which contains two distinct steps, that are iteratively executed to optimize the model parameters. It parallels the well-known expectation maximization (EM) algorithm: the first step computes the expectation over assignments $q(z)$ for each data point. Instead of computing the maximum likelihood estimate as is done in EM, in the second step we maximize the variational parameters of the posterior over model parameters. More in detail, in the first step the assignment for each data point $(s_n, c_n)$ is computed by deriving the ELBO with respect to $q(z_n)$.

$$\log q(z_n = k) = \log \gamma_n = \mathbb{E}_{q(\mu_{k,s}, \Sigma_{k,s})}[\log p(s_n | \mu_{k,s}, \Sigma_{k,s})] \tag{18}$$

$$+ \mathbb{E}_{q(\mu_{k,c}, \Sigma_{k,c})}[\log p(c_n | \mu_{k,c}, \Sigma_{k,c})] \tag{19}$$

$$+ \mathbb{E}_{q(\pi)}[\log p(\pi)] - \log Z_n, \tag{20}$$

where $Z_n$ is a normalizing term, i.e., if $\hat{\gamma}_n$ is the unnormalized logit, we can find the parameters of the categorical distribution as $\gamma_n = \frac{\hat{\gamma}_n}{\sum \hat{\gamma}_n}$.

In the second step, we compute the update of the approximate posteriors over model parameters. To this end, we derive the ELBO with respect to the natural parameters of the distributions. If the distribution is conjugate to the likelihood, we acquire updates of the following parametric form:

$$\eta_k = \eta_{0,k} + \sum_{x_n \in \mathcal{D}} \gamma_{k,n} T(x_n) \qquad (21) \qquad\qquad \nu_k = \nu_{0,k} + \sum_{x_n \in \mathcal{D}} \gamma_{k,n}, \qquad (22)$$

where $\eta$ and $\nu$ are the natural parameters of the distribution over the likelihood's natural parameters, and $T(x_n)$ are the sufficient statistics of the data $x_n$.

For the approximate posterior $q(\mu_{s,k}, \Sigma_{s,k})$ over the parameters of the spatial likelihood, the sufficient statistics are given by $T(s_n) = (s_n, s_n \cdot s_n^T)$. The NIW conjugate prior consists of a Normal distribution over the mean and an inverse Wishart distribution over the covariance matrix. Hence, for each of the prior's natural parameters, it has two values: $\eta_{0,s} = (\kappa_{0,s} \cdot m_{0,s}, V_{0,s} + \kappa_{0,s} \cdot m_{0,s} \cdot m_{0,s}^T)$ and $\nu_{0,s} = (\kappa_{0,s}, n_{0,s} + D_s + 1)$. Here, $m_{0,s}$ is the mean of the Normal distribution over the mean, $\kappa_0$ is the concentration parameter over the mean, $n_{0,s}$ indicates the degrees of freedom, $V_{0,s}$ the inverse scale matrix of the Wishart distribution, and $D_s$ the dimensionality of the MVN.

For the approximate posterior $q(\mu_{c,k}, \Sigma_{c,k})$ over the color likelihood, the sufficient statistics are again given by: $T(c_n) = (c_n, c_n \cdot c_n^T)$. The prior is parameterized similarly as the NIW over $s$: $\eta_{0,c} = (\kappa_{0,c} \cdot m_{0,c}, V_{0,c} + \kappa_{0,c} \cdot m_{0,c} \cdot m_{0,c}^T)$ and $\nu_{0,c} = (\kappa_{0,c}, n_{0,c} + D_c + 1)$. However, as we model the prior over $\Sigma_{k,c}$ as a delta distribution, we keep the values for $n_{k,c}$ and $V_{k,c}$ fixed.

Finally, the conjugate prior for the approximate posterior $q(\pi)$ over the component assignment likelihood $z$ is a Dirichlet distribution. Here, sufficient statistics are given by $T(x) = 1$, and the prior is parameterized by the natural parameter $\eta_{0,z} = \alpha$.

### 3.3 CONTINUAL UPDATES

A major benefit of VBGS is that it enables continuous learning because the parameters of each component are updated by aggregating the prior's natural parameters with the data's sufficient statistics through its assignments $q(z)$. This iterative update process is order-invariant, allowing the model to adapt without forgetting past knowledge. Components without recent data assignments revert to their prior values, preserving flexibility in the model. Crucially, assignments $q(z)$ are always computed with respect to the initial posterior over parameters $q(\mu_s, \Sigma_s, \mu_c, \Sigma_c, \pi)$. This ensures that components without prior assignments can still be used to model the data.

Concretely, when data is processed sequentially, at each time point $t$, a batch of $\mathcal{D}_t$ of data points $(s_n, c_n)$ is available, and the variational posterior over model parameters is updated. For each of these data points, the assignments $\gamma_{k,n}$ can be calculated similarly to Equation (18). We can rewrite the update steps from Equations (21) and (22), in a streaming way for $T$ timesteps:

$$\eta_k = \eta_{0,k} + \sum_{t=1}^{T} \sum_{x_n \in \mathcal{D}_t} \gamma_{k,n} T(x_n) \qquad (23) \qquad\qquad \nu_k = \nu_{0,k} + \sum_{t=1}^{T} \sum_{x_n \in \mathcal{D}_t} \gamma_{k,n} \qquad (24)$$

Hence, we can write an iterative update rule for the natural parameters at timestep $t$ as a function of the natural parameters at $t-1$:

$$\eta_{t,k} = \eta_{t-1,k} + \sum_{x_n \in \mathcal{D}_t} \gamma_{k,n} T(x_n) \qquad (25) \qquad\qquad \nu_{t,k} = \nu_{t-1,k} + \sum_{x_n \in \mathcal{D}_t} \gamma_{k,n} \qquad (26)$$

Here, $x_n$ is a placeholder for the particular data, e.g., when updating $q(\mu_{s,k}, \Sigma_{s,k})$, this would be $s_n$. At $t = 0$, the prior values for the natural parameters are used.

Note that when $\gamma_{k,n}$ is calculated using the initial parameterization of the variational posterior over parameters, applying these continual updates is identical to processing all the data in a single batch, avoiding the problem of catastrophic forgetting.

### 3.4 COMPONENT REASSIGNMENTS

In many continual learning settings, the data statistics are not known in advance. The uniform initialization of the model might not adequately cover the data space, leading to a few components aggregating all the assignments with many components remaining unused. For example, consider a room where the largest object density is at the walls, while the center mainly consists of empty space. To mitigate this problem, we introduce a heuristic for reassigning the initial location of a components to data points which the model does not explain well.

The means of $n$ components for both space and color are replaced by the location and color of $n$ data points, sampled proportional to the negative ELBO under the current model. We choose $n$ as a fraction (5%) of these unused components. A component is unused if the concentration parameters $(\alpha_k)$ of the prior over the mixture weights are equal to their prior value. By choosing $n$ as a fraction of the available components, more reassignments occur for the first frames when the object/scene is not yet known, and fewer near the end. Additionally, this ensures we do not reassign components already used at a different location. After updating the component means for these "initial" components, we can run CAVI as described in Section 3.2.

## 4 RESULTS

We evaluate our approach by benchmarking it against backpropagating gradients through a differentiable renderer. In particular, we compare the following models on the Tiny ImageNet (Le and Yang, 2015), Blender 3D (Mildenhall et al., 2020) and Habitat rooms (Savva et al., 2019) datasets:

**VBGS (Ours)**: We consider a variant of VBGS where the means of the initial posteriors $(q(\mu_s, \Sigma_s, \mu_c, \Sigma_c))$ are initialized on sampled points from the normalized data set (Data Init), as well as randomly initialized ($m_{k,s} \sim \mathcal{U}[-1, 1]$, $m_{k,c} \sim \delta(0)$) (Random Init). For models with random initialization, data normalization is performed using estimated statistics; see Appendix C for further details.

**Gradient**: In 3DGS (Kerbl et al., 2023), the parameters are directly optimized using stochastic gradient descent on a weighted image reconstruction loss $((1 - \lambda) \cdot \text{MSE} + \lambda \cdot \text{SSIM})$. In the case of image data, $\lambda$ is set to 0; in the case of 3D data $\lambda$, it is set to $0.2$. We use spherical harmonics with no degrees of freedom, i.e., the specular reflections are not modeled. In order to be able to compare performance w.r.t. model size, we also do not do densification or shrinking (Kerbl et al., 2023) (for these results see Appendix D.2). When optimizing for images, we use a fixed camera pose at identity and keep the z-coordinate of the Gaussians fixed at a value of 1. Similar to the VBGS approach, we also consider a variant where the means of the Gaussian components are randomly initialized (Random Init), and on sampled points from the dataset (Data Init).

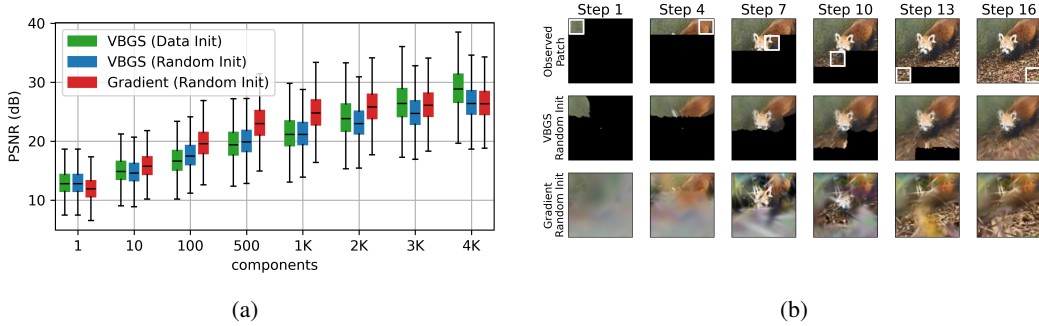

(a)  (b)

Figure 2: **Mixture model performance on image data.** (a) Shows the reconstruction performance in PSNR (dB) for various numbers of components. (b) Shows reconstruction performance in PSNR (dB) at various stages in a continual learning setting for a random initialized model (both VBGS and Gradient-based) with 1K components.

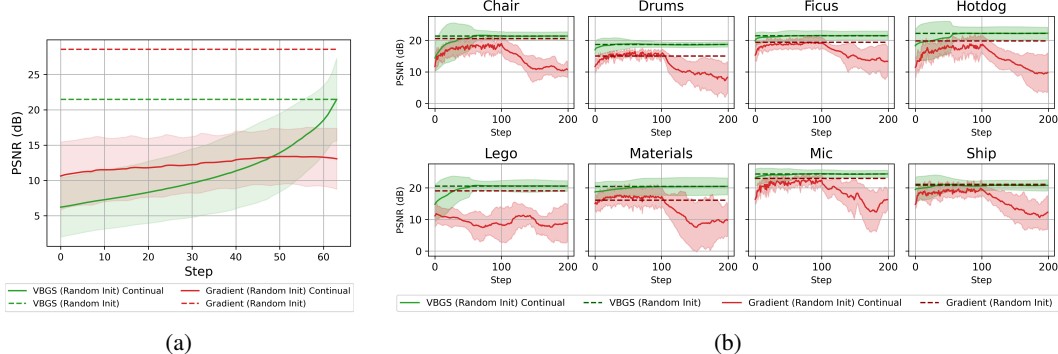

Figure 3: **Continual learning performance.** (a) Evolution of image reconstruction performance measured as PSNR (dB) after feeding image patches of size 8x8 sequentially to the model. Confidence intervals are the $Z_{95}$ interval computed over the 10k validation images of the Tiny ImageNet dataset of size 64x64. (b) Evolution of the reconstruction performance, measured as PSNR (dB), after feeding in consecutive images of an object. The shaded area indicates the 95% confidence interval computed over the 100 frames from the validation set.

### 4.1 IMAGES

We first evaluate performance on the Tiny ImageNet test set consisting of 10k images. We measure reconstruction accuracy using the Peak Signal-to-Noise Ratio (PSNR) metric, which expresses pixel error on a logarithmic scale (dB) . Reconstruction accuracy is evaluated across varying numbers of components (Figure 2a). Our results indicate that VBGS achieves reconstruction errors comparable to the gradient-based approach. We then evaluate the models in a continual learning setting, where data patches are sequentially observed and processed (Figure 2b). While VBGS maintains consistent reconstruction quality across all observed patches, the gradient-based method disproportionately focuses on the most recent patch. The PSNR over timesteps is shown in Figure 3a. VBGS converges to the same accuracy as training on the static dataset, whereas Gradient degrades due to catastrophic forgetting.

We further compared the computational efficiency by measuring the wall-clock time required for the Gradient approach to reach the performance level of VBGS after a single update step. This was computed over all images of the Tiny ImageNet validation set. We observe that VBGS is significantly (t-test, $p = 0$) faster in wall clock time ($0.03 \pm 0.03$ seconds) compared to Gradient ($0.05 \pm 0.02$ seconds).

To assess performance in a continual learning setting, we divide each image into 8x8 patches, feeding the model one patch at a time (Figure 2b). VBGS performs a single update per patch, whereas the Gradient approach performs 100 training steps per patch with a learning rate of 0.1. Figure 3a shows how the reconstruction PSNR evolves at each time step over the tiny ImageNet test set for the model with capacity 10K components. It's important to note that the reconstruction error of VBGS after observing all the patches converges to the same value as observing the whole image at once, as the inferred posterior ends up being identical. Even though the gradient approach, when observing all data at once, achieves much higher PSNR, when the data is fed in continuously, it does not achieve that performance in the continual setting, as it always focuses on the latest observed patch (see Figure 2b).

### 4.2 BLENDER 3D OBJECTS

Next, we evaluate VBGS on 3D objects from the Blender dataset (Mildenhall et al., 2020). Model parameters are inferred using 200 frames which include depth information. VBGS is trained on the 3D point cloud, which is acquired by transforming the RGBD frame to a shared reference frame. In contrast, the gradient-based approach is optimized using multi-view image reconstruction. We evaluate reconstruction performance on 100 frames from the validation set. The results, measured as PSNR, for a model with a capacity of 100K components are shown in Table 1. Our results show improved performance for both methods when components are initialized from data rather than randomly. VBGS achieves performance similar to the gradient-based approach under Data Init

Table 1: **Prediction performance for the 3D dataset**. Measured as PSNR (dB). Values ($\mu \pm \sigma$) are computed over 100 validation frames for each of the 8 blender objects. All models in this table have 100K components. The best performance for each column is marked in bold.

| | chair | drums | ficus | hotdog | lego | materials | mic | ship |
|---|---|---|---|---|---|---|---|---|
| VBGS (Data Init) | 22.82 ±0.94 | **19.50** ±0.47 | **22.06** ±0.79 | **23.62** ±1.23 | **22.53** ±0.89 | **20.55** ±1.64 | 24.78 ±0.60 | 21.23 ±0.64 |
| VBGS (Random Init) | 21.35 ±0.69 | 18.71 ±0.43 | 21.49 ±0.75 | 22.24 ±0.97 | 20.59 ±0.86 | 20.47 ±1.36 | 24.42 ±0.58 | 20.80 ±0.90 |
| Gradient (Data Init) | **22.98** ±1.23 | 19.05 ±0.57 | 21.08 ±0.92 | 21.47 ±1.67 | 19.97 ±2.24 | 20.53 ±1.64 | **25.25** ±0.90 | **23.55** ±1.20 |
| Gradient (Random Init) | 20.59 ±0.85 | 15.04 ±1.13 | 19.41 ±0.90 | 19.81 ±1.86 | 19.10 ±1.02 | 16.11 ±1.45 | 23.03 ±0.72 | 21.15 ±0.82 |

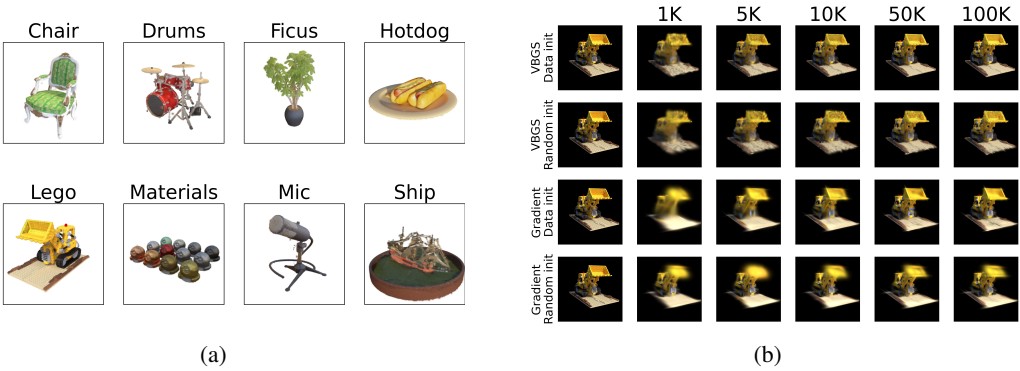

(a)                                             (b)

Figure 4: **Mixture model performance on 3D data.** (a) Image reconstructions for each of the eight objects, given a VBGS with 100k components. (b) Qualitative performance when using various amounts of components for reconstructing "lego" for both VBGS and the gradient approach.

conditions. In random initialization, our approach outperforms the gradient-based approach except for the "ship" object. For a more in-depth analysis where we vary the number of components, see Appendix D.1.

Novel view predictions for the eight blender objects are shown in Figure 4a. These renders are generated using a VBGS with 100K components, observed from a camera pose selected from the validation set. Note that these are rendered on a white background, and only the 3D object is modeled by the VBGS. Figure 4b shows the reconstruction performance as a function of the number of available components. It can be observed that for lower component regimes, VBGS renders patches of ellipsoids, while the gradient approach fills the areas more easily. We attribute this to a strong prior over the covariance shape encoded in the Wishart hyperparameters.

Finally, we also conduct the continual learning experiment for 3D and observe that the same properties from the 2D experiment hold, reaching an average reconstruction error over all objects of $11.19 \pm 3.53$ dB for VBGS (Random Init) and $21.26 \pm 1.76$ dB for Gradient (Random Init). Crucially, in the continual learning setting, the model does not have access to the data for initialization and has to be initialized randomly.

For the continual setting, 200 images are streamed in a continuous stream to the model, similar to the patches in the image experiment. For each observed (RGBD) image, VBGS applies a single update, and the gradient baseline applies 100 gradient steps with a learning rate of 0.1. We evaluate performance as the measured PSNR on novel-view prediction. Figure 3b shows the evolution of performance as a function of observed data. Note how the gradient approach's performance deteriorates after observing more frames. This figure also indicates how VBGS converges quickly to an adequate level of performance after 50 steps.

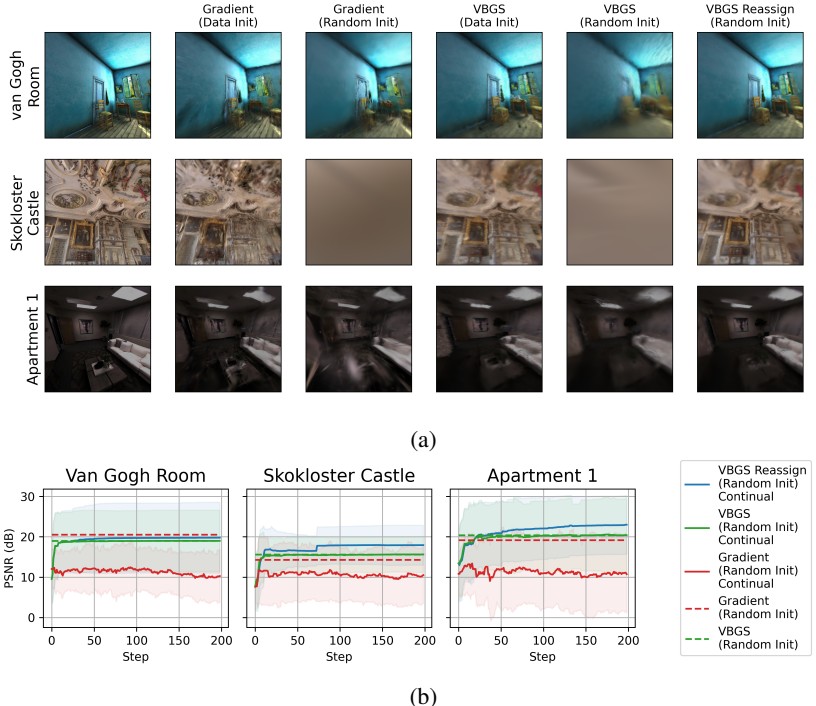

(a)

(b)

Figure 5: **Mixture model performance on 3D rooms data.** (a) Qualitative results for the different models after observing all the data. (b) Evolution of the reconstruction performance, measured as PSNR (dB), after feeding in consecutive images of the room. The shaded area indicates the 95% confidence interval computed over the 100 frames from the validation set.

### 4.3 HABITAT ROOMS

Finally, we evaluate on a dataset of 3D rooms, which is relevant for e.g. autonomous navigation for robotics in unseen environments. We consider three rooms from the Habitat test suite ("Van Gogh Room", "Apartment 1" and "Skokloster Castle") (Savva et al., 2019). The model parameters are inferred using 200 randomly sampled views from each room using the preprocessing pipeline of (Wang et al., 2023). On top of the models from the previous sections, we now also evaluate the model with component reassignments as detailed in Section 3.4.

Figure 5a shows the qualitative performance for the various considered models. Similar to previous experiments, we observe that initializing using the data yields the best performance. In contrast to the earlier experiments, random initialization does not capture well the structure of the room. This is mainly due to the randomly sampled initial means of the mixture model not covering the data well. For the Gradient approach, there are too few components that provide a gradient signal to optimize the observed view. In contrast, for VBGS, a single component might always be closer to the data than others, aggregating all the assignments. This is further evidenced by the noticeable performance improvement when using the reassign mechanism on top of VBGS. Quantitative results for all models can be found in appendix D.3

We also evaluate the models in a continual setting in Figure 5b, where 200 frames (RBGD + pose) of the environment are observed sequentially. Note that these frames are randomly sampled from the environment and are not captured using a trajectory through the room. We observe that the gradient approach does not integrate the information well, even though the views are randomly sampled and cover the room. VBGS, without reassignments, does integrate the information and reaches performance on par with the Gradient trained in the non-continual setting. Finally, we observe that adding reassignments drastically improves performance on all three rooms.

## 5  DISCUSSION AND CONCLUSION

In this paper, we introduced a novel approach to optimizing Gaussian splats using variational Bayes. Our approach achieves comparable performance to backpropagation-based methods on 2D and 3D datasets, while offering key advantages for scenarios involving continual learning from streaming data.

One limitation of VBGS compared to 3D Gaussian Splatting (3DGS) is its reliance on RGBD data, as opposed to optimizing directly on RGB projections. For many use cases we envision, such as robot navigation, depth information is often readily available from stereo vision, lidar or other sensing technologies. Furthermore, most 3DGS approaches still rely on pre-computed camera parameters obtained from structure-from-motion algorithms (Özyeşil et al., 2017), which implicitly also provide depth information through triangulation. One could also use a pretrained neural network that predicts depth from monocular RGB data (Yang et al., 2024b), as evidenced by Fu et al. (2024).

Additionally, 3DGS dynamically adjusts the model size by growing and shrinking as needed, a feature that we have not yet incorporated. In future work, we aim to explore principled approaches for dynamic model sizing, leveraging model evidence (Friston et al., 2023) to guide this process.

As VBGS represents a distribution over the parameters of a Gaussian Mixture Model (GMM) rather than optimizing for the maximum likelihood solution directly, this increases the memory requirements by a factor of 2 compared to the gradient baseline. This richer parameterization allows training to be reformulated as coordinate ascent variational inference (CAVI). The computational complexity of CAVI breaks down into computing assignments ($O(nm)$) and calculating parameter updates ($O(n)$), where n is the number of data points and m the number of Gaussian components. When all data fits in memory, a single VBGS update step was found to be faster compared to the accumulated gradient update steps to reach the same performance. However, for large-scale 3D datasets, batched processing becomes necessary, losing this competitive advantage. For a detailed discussion, please see Appendix E.

However, in streaming or partial data settings, VBGS excels by allowing updates without replay buffers, opening avenues for active data selection, a potential future research direction. Investigating active learning mechanisms, particularly in robotic SLAM or autonomous systems, could optimize data usage, ensuring that only the most informative observations are processed.

The continual learning capabilities of VBGS make it especially suited for applications requiring real-time adaptation and learning, such as autonomous navigation, augmented reality, and robotics. By continuously integrating new information without the need for replay buffers, VBGS reduces the computational burden typically associated with processing large-scale data streams. Furthermore, the variational posterior over model parameters enables parameter-based exploration (Schwartenbeck et al., 2013), allowing agents to efficiently explore and build a model of their environment in real-time. This suggests exciting opportunities for VBGS to contribute to adaptive, real-time learning systems in various real-world domains.

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

## A HYPERPARAMETERS

### A.1 PRIOR PARAMETERS

The considered conjugate priors over the likelihood parameters are parameterized by the canonical parameters shown in Table 2. Some of these values are a function of the number of available components, indicated by nc.

Table 2: **Parameters of the conjugate priors over likelihood parameters.** Some values are a function of the number of components, indicated by nc, $I$ represents the identity matrix of size the multivariate dimension $D$. Parameters are in the canonical form of the corresponding distribution.

|  |  | **2D** | **3D** |
|---|---|---|---|
| $p(\mu_{k,s}, \Sigma_{k,s})$ | $m_{s,k}$ | $\mathbf{0}$ | $\mathbf{0}$ |
|  | $\kappa_{s,k}$ | $10^{-2} \cdot \mathbf{1}$ | $10^{-2} \cdot \mathbf{1}$ |
|  | $V_{s,k}$ | $2.25 \cdot 10^4 \cdot \text{nc} \cdot I$ | $2.25 \cdot 10^6 \cdot \text{nc} \cdot I$ |
|  | $n_{s,k}$ | 4 | 5 |
| $p(\mu_{k,c}, \Sigma_{k,c})$ | $m_{c,k}$ | $\mathbf{0}$ | $\mathbf{0}$ |
|  | $\kappa_{c,k}$ | $10^{-2} \cdot \mathbf{1}$ | $10^{-2} \cdot \mathbf{1}$ |
|  | $V_{c,k}$ | $10^6 \cdot I$ | $10^8 \cdot I$ |
|  | $n_{c,k}$ | 5 | 5 |
| $p(\pi)$ | $\alpha_k$ | $\frac{1}{\text{nc}}$ | $\frac{1}{\text{nc}}$ |

### A.2 INITIAL PARAMETERS

In the first step of the coordinate ascent algorithm, $q(z)$ is inferred with an initial configuration of $q(z, \mu_s, \Sigma_s, \mu_c, \Sigma_c, \pi)$. Crucially, this configuration is distinct from the prior described in Appendix A.1. The initial canonical parameters are shown in Table 3.

The values of $m_{s,k\text{init}}$ and $m_{c,k\text{init}}$ vary for the two considered cases. When the means are initialized on the data point (Data Init), $m_{s,k\text{init}}$ and $m_{c,k\text{init}}$ are set to $K$ values $(s_n, c_n)$ sampled from the data $\mathcal{D}$. When the means are randomly initialized (Random Init), then $m_{k,s,\text{init}} \sim \mathcal{U}[-1,1]$, $m_{k,c,\text{init}} \sim \delta(0)$).

Table 3: **Parameters of the initial approximate posteriors over likelihood parameters.** Some values are a function of the number of components, indicated by nc, $I$ represents the identity matrix of size the multivariate dimension $D$. Parameters are in the canonical form of the corresponding distribution.

|  |  | **2D** | **3D** |
|---|---|---|---|
| $q(\mu_{k,s}, \Sigma_{k,s})$ | $m_{s,k}$ | $\mathbf{m}_{s,k,\text{init}}$ | $\mathbf{m}_{s,k,\text{init}}$ |
|  | $\kappa_{s,k}$ | $10^{-5} \cdot \mathbf{1}$ | $10^{-6} \cdot \mathbf{1}$ |
|  | $V_{s,k}$ | $2.25 \cdot 10^4 \cdot \text{nc} \cdot I$ | $2.25 \cdot 10^6 \cdot \text{nc} \cdot I$ |
|  | $n_{s,k}$ | 4 | 5 |
| $q(\mu_{k,c}, \Sigma_{k,c})$ | $m_{s,k}$ | $\mathbf{m}_{c,k,\text{init}}$ | $\mathbf{m}_{c,k,\text{init}}$ |
|  | $\kappa_{s,k}$ | $10^{-2} \cdot \mathbf{1}$ | $10^{-2} \cdot \mathbf{1}$ |
|  | $V_{c,k}$ | $10^6 \cdot I$ | $10^8 \cdot I$ |
|  | $n_{c,k}$ | 5 | 5 |
| $q(\pi)$ | $\alpha_k$ | $\frac{1}{\text{nc}}$ | $\frac{1}{\text{nc}}$ |

## B IMAGE RENDERING

Rendering is the process of generating an image, given an internal representation. Typically, this refers to the process of projecting from a 3D representation to the image plane. In our generative

model, this process boils down to evaluating the expected color value for each considered pixel, i.e. $\mathbb{E}_{p(c|s)}[c]$. This is straightforward to compute for image data:

$$\mathbb{E}_{p(c|s)}[c] = \sum_k \left( p(z_n = k|s) \sum_{c_i} c_i \underbrace{p(c_i|z_n = k)}_{\approx \delta(c_k)} \right) \approx \sum_k p(z_n = k|s)c_k, \qquad (27)$$

where we approximate the distribution over the color features by a delta distribution positioned at the mean of $q(\mu_c)$.

In equation (27), the distribution is conditioned on the spatial location of the pixel. For rendering in 3D, we leverage the computationally efficient 3D renderer designed by Kerbl et al. (2023). Here, the 3D Gaussians are first projected to the image plane, and by using alpha blending along a casted ray, color values are combined into a pixel color. As we do not optimize on image reconstruction, our approach does not have sensible alpha blending. We, therefore, set the alpha value for all Gaussians at 1, i.e., all components are opaque.

## C  DATA NORMALIZATION

Before training, the data is normalized to have zero mean and a standard deviation of one. This is not strictly necessary but ensures that we can use the same hyperparameters and initial parameters for all models. When all data is readily available, the data is simply normalized using the statistics from the data itself. Note that each dimension is considered individually.

When the data is not available, we assume that the random variable is distributed uniformly within a certain range: $x \sim \mathcal{U}(r_{\min}, r_{\max})$. The normalized value is then calculated as:

$$\hat{x} = \frac{x - \mathbb{E}[x]}{\sqrt{\mathrm{Var}[x]}}, \qquad (28)$$

for which the ranges for each of the parameters are displayed in Table 4.

Table 4: **Range for data normalization.** The subscript i indicates a single dimension of the vector.

|  | **Range 2D** | **Range 3D Objects** | **Range 3D Rooms** |
| --- | --- | --- | --- |
| $s_i$ | $[0, 64]$ | $[-1, 1]$ | $[-5, 5]$ |
| $c_i$ | $[0, 255]$ | $[0, 255]$ | $[0, 255]$ |

## D  ADDITIONAL RESULTS

### D.1  RECONSTRUCTION PERFORMANCE AS A FUNCTION OF NUMBER OF COMPONENTS

Figure 6a shows the reconstruction of an image from the TinyImageNet dataset using various model sizes for both VBGS and the gradient model. Notice how in low component regimes, the initialization on data yields smoother Gaussian components (e.g., in the 100 components regime); however when the capacity increases, this impact is drastically reduced. In high component regimes, the gradient approach yields a smoothed version of the surface, while the VBGS models better capture the high-frequency textures.

We evaluated reconstruction performance on the Blender 3D models for a variety of models, as a function of model size, measured as the amount of available components. The results are shown in Figure 6b.

### D.2  GROWING AND SHRINKING OF 3D GAUSSIAN SPLATTING

3D Gaussian Splatting, in its default implementation, dynamically grows and shrinks the model. In this paper, we kept the number of components fixed to make a comparison as a function of model size. Here, we optimize 3D Gaussian Splats using the gradient-based approach with dynamic model

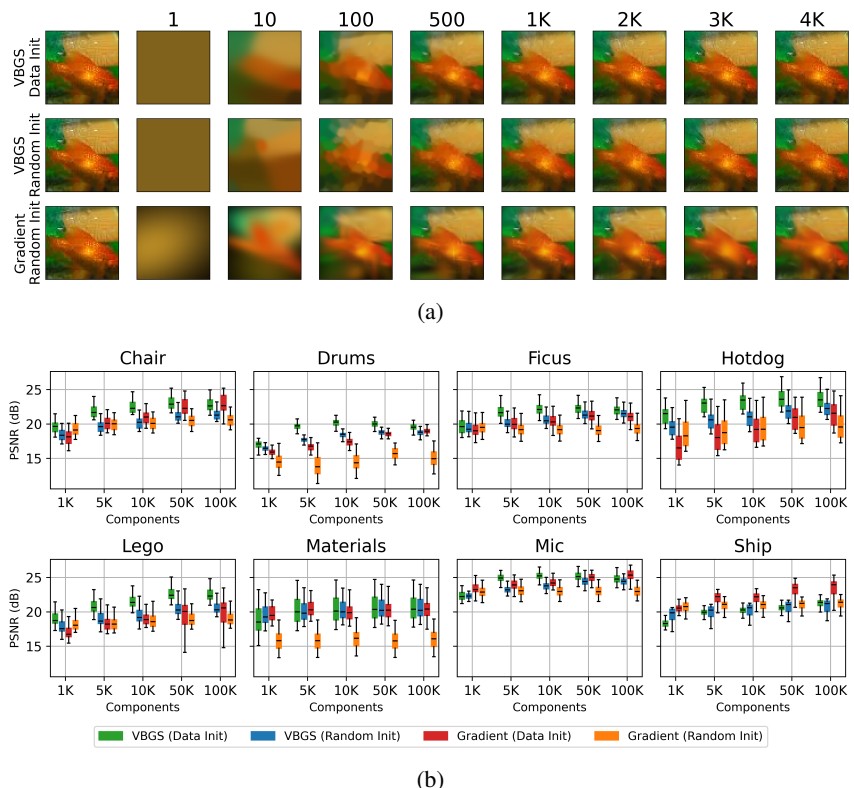

(a)

(b)

Figure 6: **Additional results for reconstruction performance**. (a) Qualitative results for image reconstruction as a function of model size. (b) Reconstruction performance for 3D models as a function of model size. Evaluated on the validation set for the 8 objects from the Blender dataset (Mildenhall et al., 2020).

Table 5: **Dynamic 3DGS.** The first row shows reconstruction performance, measured in PSNR (dB), evaluated on the validation set of the eight objects of the Blender set. The second row for each model shows the resulting amount of components after optimization. The top rows show the performance of dynamically growing 3DGS, while the bottom is the performance of a VBGS initialzied on the equivalent amount of components.

|  | chair | drums | ficus | hotdog | lego | materials | mic | ship |
|---|---|---|---|---|---|---|---|---|
| Gradient (Data Init) | 26.73 ±1.15 | 20.65 ±0.71 | 22.93 ±1.09 | 27.92 ±0.84 | 26.94 ±1.10 | 17.02 ±1.48 | 28.10 ±0.81 | 25.91 ±1.62 |
|  | 456K | 378K | 281K | 177K | 337K | 46K | 186K | 263K |
| Gradient (Random Init) | 26.72 ±1.16 | 20.65 ±0.69 | 22.94 ±1.10 | 28.06 ±0.80 | 26.98 ±1.11 | 17.03 ±1.48 | 28.26 ±0.79 | 25.81 ±1.59 |
|  | 455K | 384K | 282K | 177K | 339K | 53K | 182K | 262K |
| VBGS (Data Init) | 22.66 ±0.93 | 19.37 ±0.47 | 22.43 ±0.77 | 25.13 ±0.99 | 23.02 ±0.84 | 21.12 ±1.57 | 25.18 ±0.63 | 22.51 ±0.86 |
|  | 281K /445K | 242K /384K | 178K /282K | 111K /177K | 214K /339K | 33K /53K | 114K /182K | 165K /262K |

sizes. The results are reported in Table 5. We can see that the reconstruction quality achieved is much higher, but they also require a larger number of components.

Table 6: **Performance for novel view prediction on the Habitat rooms**. Measured as PSNR (dB). Values ($\mu \pm \sigma$) are computed over 100 validation frames for each of the three rooms. All models in this table have 100K components. The best performance for each column is marked in bold.

| | Van Gogh Room | Skokloster Castle | Apartment 1 |
|---|---|---|---|
| VBGS (Data Init) | 19.83 ±4.56 | **18.66** ±2.02 | 22.92 ±3.33 |
| VBGS (Random Init - Reassign) | 19.78 ±4.49 | 17.93 ±2.52 | 22.98 ±3.72 |
| VBGS (Random Init) | 18.98 ±3.89 | 15.62 ±2.27 | 20.38 ±4.53 |
| Gradient (Data Init) | **20.97** ±4.29 | 18.45 ±1.89 | **24.96** ±6.17 |
| Gradient (Random Init) | 20.55 ±4.02 | 14.30 ±4.36 | 19.18 ±5.32 |

Additionally, we initialize the VBGS approach on the amount of components 3DGS converges on. We observe that the performance achieved by VBGS is similar to the 100K model (see Table 1). Note that the objective of VBGS trades off accuracy with complexity, and hence a large amount of components remain unused.

### D.3 NOVEL VIEW PREDICTION PERFORMANCE ON HABITAT ROOMS

The results for the various model configurations with 100K components on the Habitat rooms data is shown in Table 7. We observe that for the random initial data condition (Random Init), VBGS performs better than Gradient for the "Skokloster Castle" and "Apartment 1", while Gradient performs better for the "Van Gogh Room". Using component reassignments, the VBGS performance can be improved even more. In the case of initialization on data (Data Init), the Gradient approach performs better on the "Van Gogh Room" and "Apartment 1".

### D.4 CONTINUAL LEARNING ON THE REPLICA DATASET

We evaluate the performance on the Replica dataset (Straub et al., 2019), which is often used to benchmark robotic SLAM systems. We train a model using a reassign fraction of $0.01$, which we found to work well for this dataset.

The sequences of the Replica datasets consist of 2000 frames. We train our model on every 10th frame, starting by frame 0 (resulting in 200 train frames), and evaluate on every 20th frame, starting by frame 5 (resulting in 100 evaluation frames).

In Figure 7a, the evolution of PSNR after observing multiple frames in a moving trajectory can be observed. Note that the PSNR steadily increases as larger aspects of the room are observed (e.g. around step 75 of Room 0, the camera turns around and observes the other side of the room).

We show qualitative results on two frames of the validation set for each of the three rooms. Note that at first some parts of the room are not yet observed and it can not generate the view. However, for each of the datasets, after frame 100 the entire room has been observed and can be accurately reconstructed without sacrificing reconstruction performance on the first observed frame.

### D.5 DEPTH-FREE VBGS

A limitation of this approach compared to traditional 3D Gaussian Splatting is that it requires depth information to be present for every frame. In this section, we run an ablation where we do not have access to depth information, but instead use a pretrained neural network (Depth-Anything-V2-Large (Yang et al., 2024a), trained on the Hypersim dataset (Roberts et al., 2021)) to estimate the metric depth of each frame.

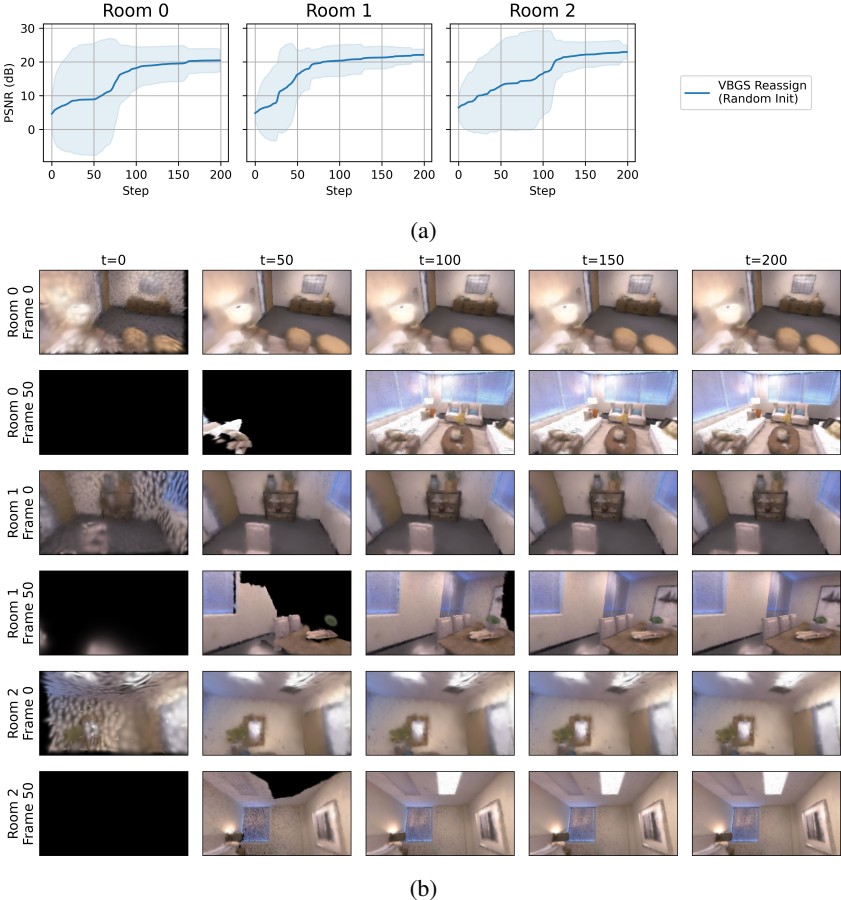

(a)

(b)

Figure 7: **Continual learning results for the Replica rooms dataset**. (a) Evolution of the reconstruction performance, measured as PSNR (dB), after feeding in consecutive images of the room. The shaded area indicates the 95% confidence interval computed over the 100 frames from the validation set. (b) Qualitative results of novel view prediction as a function of time for the three evaluated rooms.

We test this approach on "Room 0" of the Replica dataset (Straub et al., 2019), using the same train and evaluation sequence used in Appendix D.4.

As we evaluate the model on a different dataset than it was trained on, we need to calibrate the depth estimation network. We do this using ten ground truth depth frames for which we estimate the scale using linear regression without intercept, and find that for this dataset the output of the network should be scaled with a factor of 0.7.

Qualitative examples of reconstruction performance using the predicted depth model are shown in Figure 8a. Comparing these to the results in Figure 7b, we observe that the inaccuracies in the predicted depth result in larger representations of the objects (i.e. the chairs in frame 50 are now entirely covered by the flowers).

## D.6 ROBOT APPLICATIONS

We conducted a preliminary experiment using a mobile robot in the Habitat simulator. The robot observes RGBD and its pose at every frame and updates the model at each step. This is visualized in Figure 9a. The figure shows a top-down projection of the means of the Gaussians. The structure of the environment is captured without forgetting the parts observed at the beginning of the sequence. This could, for example, be used for autonomous navigation and collision avoidance.

Table 7: **Performance for novel view prediction on Replica Room 0 with estimated depth**. This compares VBGS with 100k components using random initialization and reassignments for both ground truth depth and estimated depth using DepthAnythingV2 (Yang et al., 2024a). Values are reported as PSNR (dB) ($\mu \pm \sigma$), for which the statistics are computed over the 100 validation frames.

|  | **Room 0** |
| --- | --- |
| VBGS (Random Init - Reassign) | 20.46 $\pm 1.69$ |
| VBGS (Random Init - Reassign) Depth Estimated | 16.26 $\pm 0.98$ |

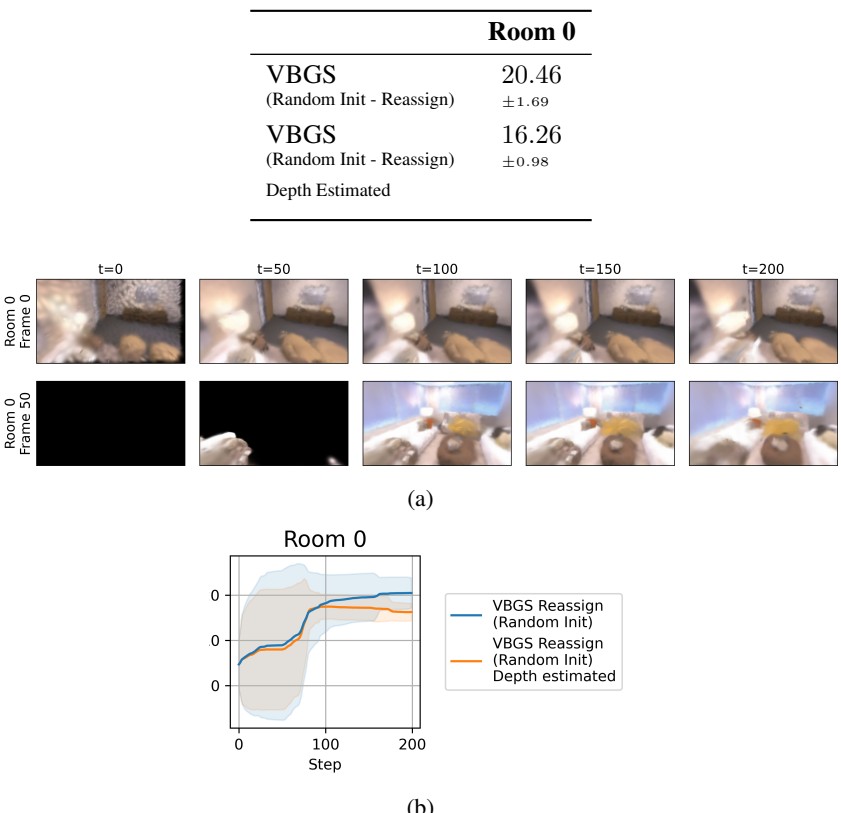

(a)

(b)

Figure 8: **Continual learning results for the Replica rooms dataset with estimated depth**. (a) Qualitative results of novel view prediction as a function of time for "Room 0" where the depth information is inferred using DepthAnythingV2 (Yang et al., 2024a). (b) Evolution of the reconstruction performance, measured as PSNR (dB), after feeding in consecutive images of the room in the case where depth information is available, as well as were the estimated depth information is used.

Additionally, we fitted a VBGS on the TUM dataset (Engelhard et al., 2011), a large-scale real-world dataset commonly used for evaluating SLAM algorithms. We train on 225 frames sampled equidistant in time across the whole sequence. Qualitative reconstruction results from the validation set are shown in Figure 9b. It can be observed that the model can capture the structure of both sides of the table.

# E   COMPUTATIONAL COST

Instead of optimizing for the maximum likelihood parameters of a Gaussian Mixture Model directly, VBGS represents a distribution over these parameters. This means that instead of representing each Gaussian component with its color, mean, scale, rotation and opacity (14 parameters per component) as is done in the gradient baseline, we represent the parameters of the Normal-Inverse-Wishart likelihood for color, position and Dirichlet prior (29 parameters per component), which increases the memory requirements by a factor of 2 for the same number of Gaussians.

This representation allows us to turn "training" into a coordinate ascent variational inference (CAVI) algorithm, which can be decomposed in two steps: computing model assignments which has complex-

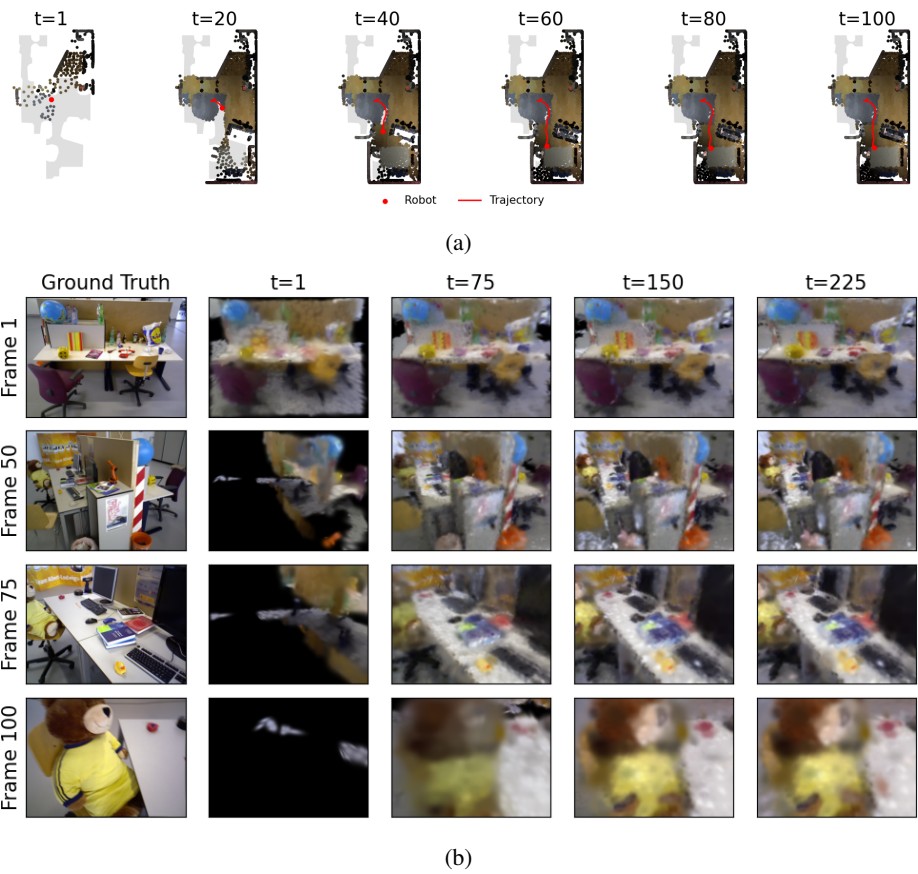

(a)

(b)

Figure 9: **Robot scenarios using VBGS:** (a) Continual mapping of an environment in the Habitat simulator (Savva et al., 2019) using a VBGS with 10K components with reassign (fraction=0.05). Shown is a top-down representation of the colored means of VBGS, where components with a z-value larger than 2m (the ceiling) are filtered out. The robot trajectory is marked in red, and the shaded gray background marks the ground truth navigable area. (b) VBGS (100K components, random init with reassign) using continual data on the TUM (Engelhard et al., 2011) dataset commonly used to evaluate SLAM algorithms.

ity O(nm) with n the number of data points and m the number of Gaussian components, and computing the parameter updates with complexity O(n). Note that we can parallelize these computations over both the number of components as well as data on a GPU.

We conducted an experiment where we compare the computational cost of a single update step with the Gradient approach. Specifically, we first execute a single VBGS update, and then train the gradient-based Gaussian splat until it reaches the same level of reconstruction performance as VBGS (measured in PSNR). The difference in variance between VBGS and the gradient approach can be attributed to the fact that VBGS has a fixed number of steps (1) whereas for the gradient approach this is dependent on the reconstruction quality.

Given that all the data fits in memory, VBGS is significantly faster than the gradient baseline. However, for the 3D experiments this is no longer the case and we need to resort to batched processing of the observed data. An interesting venue for future work would be to actively sample data points and only compute the update for these points.

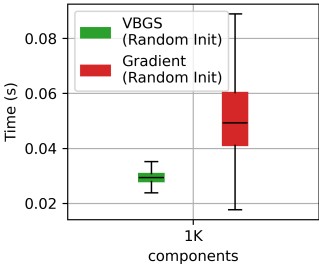

Figure 10: **Computation time for VBGS and Gradient approach**. Values are measured as wall clock time on an NVIDIA RTX 4090 in seconds over the validation frames of the ImageNet dataset. Both models have 1K components.

