# OpenReview forum: "Variational Bayes Gaussian Splatting"
_ICLR.cc/2025/Conference — Submitted to ICLR 2025_

### Official Review · Reviewer_Yuyf · 2024-10-28

**Soundness:** 1
**Presentation:** 2
**Contribution:** 1
**Rating:** 3
**Confidence:** 3

**Summary:**

This paper proposes Variational Bayes Gaussian Splatting (VBGS), which frames training a Gaussian splat as variational inference over model parameters.

**Strengths:**

1 The paper is well-structured.
2 The idea of introducing the conjugacy properties of multivariate Gaussians seems to make sense.

**Weaknesses:**

1 In the introduction, and background (e.g., 3D Gaussian Splatting) and research gap are not detailed enough, so the author's research motivation cannot be understood clearly.
2 The novelty is unclear and insufficient, it's confused why the proposed method can eliminate the need for replay buffers. More relevant discussions and technical details should be analyzed.
3 The authors claim that this work can "more robust and efficient parameter estimation." However, no computational efficiency analysis (e.g., computational time, memory usage) is provided.
4 In the experiments, the baseline only includes "Gradient". More discussions are need to explain why choosing"Gradient". Besides, the superiority of the proposed method over "Gradient" needs more analysis from both theoretical and experimental perspectives. Are there any more recent and advanced baselines?
5 The evaluation metrics should be introduced in detail, e.g., PSNR, to enhance the understandability and readability of the experimental part.

**Questions:**

Please refer to the weaknesses.

---

> ### Author Response · Authors · 2024-11-20
> **Response to review**
>
> Dear Reviewer,
>
> We appreciate your thoughtful feedback and have revised our paper to accommodate the identified weaknesses:
>
> **Introduction and background**
>
> We updated the introduction to better motivate our work and highlight the research gap and contributions.
>
> **Unclear Novelty and Elimination of Replay Buffers**
>
> The elimination of replay buffers is a direct result of VBGS’s variational Bayesian formulation, which enables parameter updates that integrate new data while retaining prior information through a variational posterior. Unlike gradient-based methods, which require access to old data to mitigate forgetting, VBGS maintains a probabilistic representation of parameters that naturally accumulates evidence over time. We updated section 3.3 to better highlight this mechanism.
>
> **Computational Efficiency Analysis**
>
> We agree that computational efficiency is a critical aspect of practical usability. We updated the discussion with a paragraph on memory usage and computational efficiency of VBGS compared to 3DGS. We also added Appendix E with a more in-depth discussion on computational cost.
>
> **Baseline Selection and Limited Comparison**
>
> The choice of the "Gradient" baseline was motivated by the direct relevance of 3DGS as a closely related state-of-the-art method for Gaussian mixture-based scene representation. Given our focus on the continual learning use case without replay buffer, we did not find any other baselines suited for an apples-to-apples comparison. However, we did expand our benchmarks to include the Replica dataset (Straub et al, 2019), as employed in the SLAM benchmark of Keetha et al., where we feed camera frames from a particular SLAM trajectory in-order and do continual updates on VBGS.
>
> **Evaluation metrics**
>
> Thanks for pointing that out. We updated the manuscript to introduce PSNR as the metric for reconstruction quality when it first appears in the evaluation section.
>
> We hope these revisions address the main concerns raised.

---

> > ### Comment · Reviewer_Yuyf · 2024-11-25
> >
> > Dear Authors:
> >
> > Thanks for the rebuttal. Considering the technical quality, presentation, and novelty, I decided to maintain my current score and I will keep discussion with other reviewers in the future.

---

> > > ### Author Response · Authors · 2024-11-25
> > >
> > > Dear reviewer,
> > >
> > > Thank you for checking in and taking note of our rebuttal. Should you have any recommendations for us to further improve our work, that would be greatly appreciated.

---

### Official Review · Reviewer_f4Be · 2024-11-02

**Soundness:** 2
**Presentation:** 3
**Contribution:** 2
**Rating:** 5
**Confidence:** 3

**Summary:**

This paper introduces Variational Bayes Gaussian Splatting (VBGS), a new method for modeling 3D scenes with Gaussian splats that addresses the challenge of catastrophic forgetting in continuous data streams. Unlike traditional methods that rely on backpropagation through a differentiable renderer and require replay buffers, VBGS formulates training as variational inference over model parameters, enabling efficient updates through a closed-form variational update rule.

**Strengths:**

1. Effectively handles continuous data streams: By framing the training process as variational inference over model parameters, VBGS avoids the issue of catastrophic forgetting typically seen in continuous data streams with traditional methods.

2. No need for replay buffers: VBGS leverages the conjugacy properties of multivariate Gaussians, deriving a closed-form variational update rule that allows efficient incremental updates without requiring replay buffers.

**Weaknesses:**

1. Limited improvement. In Tab. 1, your method is comparable to the Gradient method (3DGS), which cannot prove the effectiveness of your method.

2. Unlike 3D Gaussian Splatting (3DGS), it relies on RGBD data. It means your method needs a depth camera and does not outperform 3DGS.

3. The experiments are not comprehensive, as they do not include evaluations on standard datasets like mip-NeRF 360, which could provide a more robust comparison with existing methods.

4. The comparison with baselines is limited; a more thorough evaluation would include comparisons with established methods such as Mip-NeRF 360, Instant-NGP, and NeRF. Including these methods as baselines would provide a clearer understanding of VBGS's performance relative to a broader range of state-of-the-art approaches.

**Questions:**

As stated in the weaknesses above, I have some additional questions as follows:
1. Why is it said that 3DGS struggles with catastrophic forgetting when dealing with continuous streams of data?

2. Could you provide numerical results on the HABITAT ROOMS dataset?

---

> ### Author Response · Authors · 2024-11-20
> **Response to review**
>
> Dear Reviewer,
>
> First of all, we would like to thank you for the detailed feedback and for identifying areas where our work could be clarified or strengthened.
>
> **Limited Improvement in Table 1**
>
> We acknowledge that the PSNR values in Table 1 show comparable performance to 3DGS in the static dataset setting. However, our main focus is on scenarios involving continual learning, where VBGS provides significant advantages over 3DGS due to its resilience against catastrophic forgetting. This advantage is most evident in our continual learning experiments, where 3DGS performance degrades as new data is introduced, whereas VBGS maintains consistent performance.
>
> **Reliance on RGBD Data**
>
> We recognize that reliance on RGBD data is a limitation compared to 3DGS, which directly optimizes over RGB projections. However, for the use cases we envision, i.e. robotics, 3D data is often available through RGBD or lidar sensors. In the absence of depth data, one could resort to a depth estimation method in combination with VBGS. To demonstrate this, we added an experiment to the appendix where depth information is not available, and is estimated through a pretrained neural network (using metric DepthAnythingV2).
>
> **Limited Baseline Comparisons and Datasets**
>
> We chose to limit our comparison to Gaussian splatting approaches as our method specifically proposes an alternative method for optimizing Gaussian Mixture Models with benefits for continual learning problems. This way, we are able to effectively compare gradient based Gaussian splatting with VBGS for a mixture model with N components, while for NeRFs this kind of parameterizations are hard to define.
>
> Given our focus on continual learning without a replay buffer, we did not find any other baselines suited for fair comparison. However, we did expand our benchmarks to include the Replica dataset (Straub et al, 2019), as employed in the SLAM benchmark of Keetha et al., where we feed camera frames from a particular SLAM trajectory in-order and do continual updates on VBGS.
>
> **Specific answers to questions:**
>
> **Why does 3DGS struggle with catastrophic forgetting?**
>
> Catastrophic forgetting arises in 3DGS because it relies on gradient-based optimization, which updates parameters for new data without explicitly preserving information about prior data. In streaming data settings, older data is often inaccessible or not stored in memory, leading to overwriting of previously learned features. Mitigation strategies, such as replay buffers, can alleviate this issue but require significant memory and computational resources.
> VBGS avoids catastrophic forgetting by maintaining a variational posterior over parameters, allowing updates that integrate new information while preserving knowledge of previous observations. We expanded our discussion of these mechanisms in the introduction to better contextualize this advantage.
>
> **Numerical Results on the Habitat Rooms Dataset**
>
> The numerical results on the Habitat rooms dataset can be found in Table 6 of Appendix D.3.

---

> ### Comment · Reviewer_f4Be · 2024-11-27
>
> Dear Authors,
>
> Thank you for your detailed rebuttal. After carefully considering the technical quality, presentation, and novelty of your work, I have decided to maintain my current score. I will continue discussions with the other reviewers to ensure a fair and thorough evaluation.

---

### Official Review · Reviewer_ivJV · 2024-11-04

**Soundness:** 3
**Presentation:** 2
**Contribution:** 3
**Rating:** 5
**Confidence:** 5

**Summary:**

The paper proposes a reformulation of 3D Gaussian splatting with variational inference framework. This allows a closed-form update rule and applicability to continual learning framework. The model is compared with gradient-based update rule in small-scale (~100k) 3DGS fitting to Habitat test suite and NeRF synthetic dataset and shows performance boost with better robustness to randomized initialization.

**Strengths:**

The main strengths I find in this paper lie in its novelty, clarity in writing, and reproducibility.

- Both relating 3D Gaussian splatting as Gaussian mixture model and variational inference is new to this field, and this novelty should be acknowledged.
- Application of continual learning well justifies the usage of variational inference framework.
- The training scheme is described in detail and the described motivation and methodology is quite clear.
- The supplementary material has the code implementation for reproducibility.

**Weaknesses:**

Although I find the method informative, there are several points to be clarified in order to improve the presentation.

- Common 3DGS involves several millions of Gaussians, but the presented experiments are only based on 100k Gaussians, which is very slim. This small parameter constraint makes the PSNR scores in Table 1 way below the numbers reported by other NeRF/3DGS papers which go high above 30 dB for the same NeRF synthetic dataset. Is there any specific reason (e.g., computational burden) that constraints the experiments? If so, this should be specified.
- Unlike the authors’ claim in Appendix D.3, Table 6 shows that the presented VBGS shows comparable or worse results than Gradient-based method, which needs further clarification.
- It seems like the method does not incorporate cloning/pruning/splitting of Gaussians which take crucial roles in the original 3DGS. By scaling up the Gaussians during the training, the performance can be boosted significantly, so unless there is a counterpart in VBGS for resizing the Gaussian pool, I believe there is a little advantage of using this framework in practical scenarios. I believe there should be an experiment with VBGS having initial Gaussians of the same size of the final Gaussians in Table 5, so we can compare fairly. For example, if we start from 456K Gaussians in VBGS and performs better than dynamically resized 3DGS in Table 5, we can clearly say that there is a reason to migrate from 3DGS to VBGS.
- I believe it is better to have a dedicated discussion on changes in computational demand relative to the size of Gaussians in different frameworks. Inference time should be the same, but training time and the memory requirement will be very informative to followers to this work.

In summary, unless there is a clear advantage of VBGS over standard size-varying gradient-based 3DGS, the method is not practically persuasive, yet. Furthermore, currently, there is no guarantee of scalability, which I believe important in learnable 3D representation. It is OK to have a slow and memory-intensive method if the performance (PSNR/SSIM) is stronger, but regarding Table 6, the presented method is not yet persuasive. Regarding these, the current version of the manuscript seems not ready to be published despite its novel and intuitive idea. Please note that I am flexible with my score, and looking forward to further discussion in the rebuttal phase.

**Questions:**

These are minor concerns that are not counted in my final scores.

- Having only three components in Figure 2.a without Gradient (Data init) seems incomplete. Please have it updated.
- As the authors have demonstrated with a continual learning framework, is it workable to scene in motion?
- Does this method scale up? Can this method scale up to tens of millions of Gaussian splatting for larger datasets (e.g., MipNeRF-360)?

---

> ### Author Response · Authors · 2024-11-20
> **Response to review**
>
> Dear Reviewer,
>
> Thank you for your valuable feedback on this manuscript.
>
> **Limited Number of Gaussians in Experiments, Memory Requirements and Computational Demand**
>
> It is true that the current state-of-the-art 3DGS methods have millions of components and we are only comparing in a range of thousands. This has a couple of reasons. First and foremost our focus is on continual learning without a replay buffer, and not so much on achieving better reconstruction quality. We believe that this is important for a lot of practical applications (e.g.  robot navigation). Second, our variational update scheme naturally trades off accuracy and complexity. Hence, depending on the prior, it will favor less components if the price to pay in accuracy is low. See also point 3. Third, to balance the experimentation time and compute cost, we did not focus on large-scale datasets such as MipNeRF-360.
>
> Instead of optimizing a Gaussian Mixture Model (GMM), VBGS in fact represents a distribution over GMMs and does variational inference to get a posterior distribution. This means that instead of representing each Gaussian component with its color, mean, scale, rotation and opacity (=14 parameters per component) as done in 3DGS, we represent all parameters of the Normal-Inverse-Wishart likelihood for color, position and Dirichlet prior (= 29 parameters per component), which does indeed increase the memory requirements by a factor of 2 for the same number of Gaussians.
>
> This representation allows us to turn “training” into a coordinate ascent variational inference (CAVI) algorithm, which can be decomposed in two steps: computing model assignments which has complexity O(nm) with n the number of data points and m the number of Gaussian components, and computing the parameter updates with complexity O(n). Note that we can parallelize these computations over both the number of components and the data using accelerated hardware.
>
> In terms of compute time it is hard to do an apples-to-apples comparison with 3DGS, as although 1 CAVI update step is more consuming  than 1 gradient update step, you typically need way less CAVI update steps to converge. For the image experiments, we measured the average compute time for both CAVI and gradient descent to converge to the same loss, and found that VBGS was significantly faster. However, for the larger 3D experiments, we had to split the data in several batches because of limited GPU memory on our machine, which resulted in a worse overall runtime for VBGS. An interesting venue for future work would be to actively sample data points and only compute the update for these points.
>
> We added a section in the Appendix E highlighting this aspect.
>
> **Comparison with Gradient-Based Methods (Table 6)**
>
> With respect to Table 6 in Appendix D.3., we would like to emphasize that in the case of random initialization VBGS (both with and without reassign) performs better than Gradient for “Skokloster Castle” and “Apartment 1”, while Gradient performs better for the “Van Gogh Room”. In the case of initialization on data, the Gradient approach performs better on the “Van Gogh Room” and “Apartment 1”. However, note that all these values are within the standard deviation of each other. We added a section in the appendix to describe this table in more detail.
>
> **Lack of Dynamic Gaussian Pool Resizing**
>
> We conducted the suggested experiment where we initialize VBGS with approximately the same number of components as the resulting size of a dynamically growing 3DGS. We observe that the performance of VBGS does not improve over a model with 100K components. This has two particular reasons: 1) the prior size of the components is set inversely proportional to the number of components (i.e. if more components are used, they should each be smaller). However we notice that when they are too small this . 2) The objective function (the evidence lower bound) naturally trades off complexity with accuracy and prefers models of smaller size. The updated table indicates the amount of components that are used for VBGS.
>
> **Specific answers to the questions:**
>
> Figure 2.a Missing "Gradient (Data Init)" Component
>
> Thank you for pointing this out. We will set up the data init experiment for the gradient baseline and update Fig 2.a in the coming days.
>
> **Applicability to Scenes in Motion**
>
> In its current form this model does not handle scenes in motion. To deal with moving objects, one would need to add a dynamics model that iteratively predicts the next spatial location of each of the components. An approach that could be used in combination with VBGS to deal with changing scenes is to work with a forgetting rate, where the update of new data is weighed more than the prior statistics, and hence the model can overwrite the changed parts of the scene with the new observation.
>
> We thank the reviewer for their flexibility and look forward to further discussions during the rebuttal phase.

---

> > ### Comment · Reviewer_ivJV · 2024-11-24
> >
> > Dear Authors,
> >
> > Thank you for the detailed feedback and for the update of the manuscript. I will summarize my concerns as follows:
> > - **(Regarding the Number of Gaussians in the Demonstration)**: I definitely agree with the authors on the theoretical novelty of this work using variational inference (CAVI) for training Gaussian splatting. However, 3DGS is currently an extremely application-oriented topic focusing on a narrow problem of construction of a three dimensional representation from multiple captured images. Therefore, I believe 3DGS works should at least be demonstrated with real-world datasets. If the resource constraints are preventing this demonstration, the claimed innovations are very likely to be regarded less meaningful. But I believe we have a workaround to this problem by targeting small-scale real-world examples of Gaussian splatting such as Gaussian head avatars/Gaussian human avatars which takes 10 times less Gaussians than the scenery datasets. Experiments of the current version are not yet persuasive.
> > - **(Regarding Table 6)**: If so, the table still demonstrates that the presented method is at most comparable to the gradient descent. Since VBGS seems like taking larger resources (from the answer 1), we may keep using gradient-based method than VBGS.
> > - **(Dynamic Gaussian)**: Thank you for the honest report on this. I am still very interested in the idea you proposed for training a Gaussian splatting. However, the idea seems to require more experimental proofs of effectiveness in order to be accepted as a contribution to this field.
> >
> > I will retain my score and will wait for further clarification.
> >
> > Best,
> > Reviewer ivJV

---

> > > ### Author Response · Authors · 2024-11-25
> > >
> > > Dear reviewer ivJV,
> > >
> > > We appreciate your acknowledgement of the theoretical novelty of our work. We agree that one would continue using 3DGS rather than VBGS in cases where a) you have a static train dataset and where b) maximizing PSNR is your main metric. However, VBGS is specifically targetting different use cases, particularly where you need to deal with continuously streaming data and maximizing accuracy might be not be your prime requisit. For example, a robot navigating an apartment needs to represent the 3D space in order to safely navigate and perform tasks. We believe it is not necessary to have 100M components to get a pixel-perfect representation of the painting on the wall, but it should make sure not to bounce into or damage its environment. Moreover, in this case a gradient-based optimization scheme forces one to keep a replay buffer of the ever-growing set of observed frames to avoid catastrophic forgetting, as evidenced by our experiments.
> > >
> > > To further strengthen our claim, we added more experiments showcasing how a robot builds an increasingly better representation of an apartment, as well as qualitative results on a real-world SLAM dataset (TUM) in Appendix D6.
> > >
> > > We hope this further mitigates your concerns, and thank the reviewer for the response and interaction.

---

> > > > ### Comment · Reviewer_ivJV · 2024-11-26
> > > >
> > > > Thank you for the quick comment. I understand that there can be on additional contribution of this work--for continual learning. It is indeed nice and interesting application for this algorithm. However, such contributions must be carefully proven with adequate set of experiments. There are works like CL-NeRF (continual learning NeRF) and many (at least 5) Gaussian Splatting SLAM trying to solve similar problems. The added Figure 7 in Appendix D6 is indeed interesting. But it would be much better if we have a nice comparison with existing works. I still believe this work is not perfectly prepared for demonstrating the usefulness of the nice idea you are presenting. Please understand I cannot you give a better score unless standardized set of experiments are conducted with apt comparisons with previous works. I will keep my score.

---

> > > > > ### Author Response · Authors · 2024-11-26
> > > > >
> > > > > Dear reviewer ivJV,
> > > > >
> > > > > We are aware of works like CL-NeRF (which we cite) and the Gaussian Splatting SLAM works (please let us know if we missed some). However, all of these rely on the backpropagation of gradients, and these require a replay buffer to work. In addition, these approaches benefit from standing on the shoulders of a decade of software optimizations of backpropagation frameworks. During the rebuttal period, we have added experiments on datasets that these Gaussian Splatting SLAM works use (e.g. the Replica dataset is used in (Matsuki et al), (Keitha et al), and the TUM RGB-D dataset is used in (Matsuki et al).
> > > > >
> > > > > We agree that we could run the experiments of the other authors, however, the conclusion will remain unchanged: their methods will achieve a higher PSNR at the expense of an increasing replay buffer, as we show in many experiments throughout the paper. We acknowledge that the field has shifted to an extremely application-oriented topic where the main merit of a paper is measured by reconstruction PSNR, rather than its algorithmic contribution. However, we remain confident that our approach is beneficial for certain applications, in particular, where data is streamed.

---

> > > > > > ### Comment · Reviewer_ivJV · 2024-11-26
> > > > > >
> > > > > > Dear Authors,
> > > > > >
> > > > > > I understand the authors' opinion. It will be interesting to compare this work with other Gaussian SLAM works with full scale. Please understand that I have to remain conservative to the advantage of this work in continual/streaming *applications* unless there are comprehensive quantitative comparisons at scale. I will keep my score until further verification is presented. Thank you for the active discussion regarding the review. It helped me understanding your novel idea.
> > > > > >
> > > > > > Best,
> > > > > > Reviewer ivJV

---

> > > > > > > ### Author Response · Authors · 2024-11-26
> > > > > > >
> > > > > > > Dear Reviewer,
> > > > > > >
> > > > > > > Thank you for your understanding. We agree that application-level comparisons at scale would make for a great variational Bayesian Gaussian Splatting SLAM paper. However, this was out of scope for the current work where the focus was the optimization algorithm.
> > > > > > >
> > > > > > > Thank you for actively engaging and recognizing the novelty of the idea.
> > > > > > >
> > > > > > > Kind regards

---

### Official Review · Reviewer_WzD1 · 2024-11-04

**Soundness:** 3
**Presentation:** 3
**Contribution:** 3
**Rating:** 5
**Confidence:** 4

**Summary:**

This paper proposes Variational Bayes Gaussian Splatting (VBGS) for modeling 3D scenes by framing training of Gaussian Splats as variational inference. The formulation enables closed-form variational update of model parameters which is particularly beneficial in online settings. In particular, updating the model from a sequence of observations is equivalent to processing all data in a single batch. Thus, catastrophic forgetting is circumvented.

Experimental results indicate VBGS matches the performance of 3D Gaussian Splatting (3DGS) on static datasets while enabling continual learning with streamed 3D data.

**Strengths:**

**S1.** The method addresses an important limitation of 3DGS.

**S2.** The method is principled and presented with clarity.

**S3.** The experimental validation lends support to the practical usefulness of the method: in particular, in the online setting 3DGS is clearly impacted by catastrophic forgetting while VBGS is not.

**Weaknesses:**

**W1.** The method is motivated by the application to continual learning scenarios such as SLAM. However, the experimental validation does not compare against recent SLAM methods or datasets, e.g., (Matsuki et al., 2024; Keetha et al., 2024).

**W2.** The significance of the mean-field approximation is not discussed or investigated experimentally.

**W3.** The method's reliance on depth maps is, as acknowledged by the authors, a limitation of the method. The author’s propose inferring depth maps from RGB data using a pretrained model. Experiments along these lines would have been interesting to include.

**Questions:**

**Q1:** Why not evaluate using datasets and protocols from recent SLAM works such as (Matsuki et al., 2024; Keetha et al., 2024)?

**Q2.** When should one expect the mean-field approximation to hurt performance?

**Details Of Ethics Concerns:**

NA.

---

> ### Author Response · Authors · 2024-11-20
> **Response to review**
>
> Dear Reviewer,
>
> First of all, we would like to thank you for these valuable insights and suggestions.
>
> **W1: Lack of comparison against recent SLAM methods or datasets**
>
> Thank you for raising this point. Our goal was to demonstrate VBGS as a general-purpose solution for continual learning in Gaussian splatting. To ensure focus, we selected benchmarks aligned with 2D image / 3D scene representation and online learning exploiting the conjugate properties of Gaussian Mixture Models. That said, SLAM-specific benchmarks would indeed provide additional insights into VBGS’s performance in real-world navigation tasks.
>
> We would like to emphasize that we validated our approach on 3 rooms of the Habitat dataset (Manolis, 2019). However, as the data collected is not in a continual trajectory, we now added three rooms from the Replica dataset (Straub et al, 2019), as employed in the suggested work of Keetha et al. Concretely, we feed camera frames from a particular SLAM trajectory in-order and do continual updates on VBGS. We did not integrate with camera tracking, as Keetha et al backpropagate gradients w.r.t the pose to track camera pose. Developing a pose update rule using a VBGS model is left as future work.
>
> **W2: Significance of the mean-field approximation**
>
> The mean-field approximation was adopted to ensure computational tractability while still maintaining a variational posterior over Gaussian parameters.  This approximation is crucial as this allows to update each variable independently as a separate step in the coordinate ascent algorithm. We added a sentence in the main text to emphasize this point.
>
> In our experiments, we observed that VBGS achieved competitive results compared to gradient-based methods like 3DGS, indicating that the mean-field assumption does not significantly hinder performance in the evaluated settings. However, we agree that the conditions under which this approximation might fail warrant further exploration.
>
> **W3: Reliance on depth maps**
>
> In line with your suggestion, we added an experiment to the appendix where depth information is not available, and is estimated through a pretrained neural network (using metric DepthAnythingV2). We show that we are still able to learn the room structure at the expense of a drop in PSNR (from 20.46 to 16.26).  Note that the quality of the resulting VBGS will depend heavily on the accuracy of the predicted depth.
>
> **Specific answers to questions:**
>
> Q1: As the main focus of this paper is not to propose a SLAM system (we currently do not have camera tracking) we evaluated on other datasets. However, we now added three rooms of the Replica dataset.
>
> Q2: The mean-field approximation assumes independence between parameters in the posterior, which simplifies updates but may limit flexibility when parameter dependencies are critical. We expect the performance of VBGS to degrade in scenarios where:
> The true posterior exhibits significant correlations between parameters, as these cannot be captured by the mean-field approximation.
> Highly multimodal posteriors are present, where mean-field assumptions can lead to overly simplified solutions.
> In our current experiments, the mean-field approximation performed well, likely due to the Gaussian mixture model structure, where dependencies between parameters are less pronounced, and the mixture can represent multimodal distributions.
> We are grateful for the reviewer’s thoughtful comments, and we hope to have satisfied all the reviewer's concerns and we look forward to receiving updated feedback.

---

> > ### Author Response · Authors · 2024-11-25
> >
> > Dear reviewer,
> >
> > In addition to our previous revision, we have now added extra experimental results focusing on the robot navigation use case, showing how a VBGS splat is expanded as more frames are added in the Habitat simulator, as well as qualitative results on the TUM dataset, a real-world SLAM benchmark (Engelhard, 2011). You can find these results in Appendix D.6. We hope this further strengthens our manuscript and addresses previously raised concerns.

---

> > > ### Comment · Reviewer_WzD1 · 2024-11-30
> > >
> > > I thank the authors for the additional comments and experiments. While I appreciate the additional insights, I am keeping my current score as the additional experiments are mostly qualitative. Similar to reviewer ivJV, I believe additional work is required to demonstrate the usefulness of the proposed method. In particular, a comparison with prior work on commonly used benchmarks would likely be required to increase the score above the acceptance threshold.

---

### Meta-Review · Area_Chair_ywdm · 2024-12-19

**Metareview:**

The paper proposes a variational Bayesian approach to learning Gaussian splats. The main claim is that the method matches state-of-the-art for static datasets but works much better for continual and streaming data.

The work shows promising results, especially in mitigating the 'catastrophic forgetting' encountered when running traditional Gaussian Splatting methods in a streaming setting.

However, the experimental part seems too preliminary to warrant publication at ICLR. Comparisons to baselines (as used in SLAM methods, which work in a streaming setting) are mostly missing, and the results are rather small scale.

If these aspects are improved, this work will make a strong submission. However, at this stage, I recommend to reject the paper.

**Additional Comments On Reviewer Discussion:**

All reviewers liked some aspects about the work, such as motivation, formulation and how the approach seems to work in small scale setting. However, all reviewers also pointed out problems with the experimental evaluation (missing baselines, no large scale experiments). I agree with these points, and therefore formed the decision to rejected the work.

---

### Decision · Program_Chairs · 2025-01-22

Reject